# Redundancy and dependency in brain activities

**Sikun Lin**    **Thomas Sprague**    **Ambuj K Singh**
UC Santa Barbara
{sikun,tsprague,ambuj}@ucsb.edu

## Abstract

How many signals in the brain activities can be erased before the encoded information is lost? Surprisingly, we found that both reconstruction and classification of voxel activities can still achieve relatively good performance even after losing 80%-90% of the signals. This leads to questions regarding how the brain performs encoding in such a robust manner. This paper investigates the redundancy and dependency of brain signals using two deep learning models with minimal inductive bias (linear layers). Furthermore, we explored the alignment between the brain and semantic representations, how redundancy differs for different stimuli and regions, as well as the dependency between brain voxels and regions.

## 1   Introduction

Natural images are extremely rare in the vast image space, making them highly compressible (Brunton & Kutz, 2022). Similarly, brain signals only occupy a small fraction of the signal space they reside in, and many signal dimensions are redundant–if we only care about encoding meaningful brain activities. Stevens (2015) linked neural activity modeling to compressed sensing because of its low-dimensional nature, and Rose et al. (2021) proposed decomposing signals into presentations of visual motifs as the way the brain performs compact neural encoding. With the recent emergence of larger-scale brain imaging datasets, we now have the chance to systematically study the redundancy and dependency of brain signals with deep learning models. In this work, we first examine the redundancy of brain signals in two tasks: signal reconstruction, in which the goal is to reproduce the activities across all voxels using only a subset of voxels, and classification, in which the goal is to determine the visual categories that appear in the stimuli. We embed the brain signals into a latent space and analyze the number of latent dimensions and interpolations. We also study the redundancies and dependencies in different visual subtasks, hemispheres, regions, and voxels. These steps help us analyze the dependencies and redundancies across different parts of the brain and reveal important insights about the visual brain architecture and representations.

## 2   Dataset and models

We conduct our studies on the Natural Scenes Dataset (NSD) (Allen et al., 2022), which records functional magnetic resonance imaging (fMRI) signals while the subjects view natural images. It provides high-resolution scans with a high signal-to-noise ratio (SNR) at an unprecedented scale. The number of samples in NSD allows better training and investigation of brain data using deep learning models. In particular, we directly utilize the provided fMRI betas, the estimated response amplitude of each voxel to each trial, obtained through the general linear model (GLM). There are eight subjects in NSD, and the following results are from subjects 1, 2, and 5: 1 is the principal subject, and 2, 5 are used for verification (result plots correspond to subject 1, if unspecified). We focus the study on one region of interest (ROI), nsdgeneral, which covers voxels responsive to the NSD experiment in the posterior aspect of the cortex.

4th Workshop on Shared Visual Representations in Human and Machine Visual Intelligence (SVRHM) at the Neural Information Processing Systems (NeurIPS) conference 2022. New Orleans.

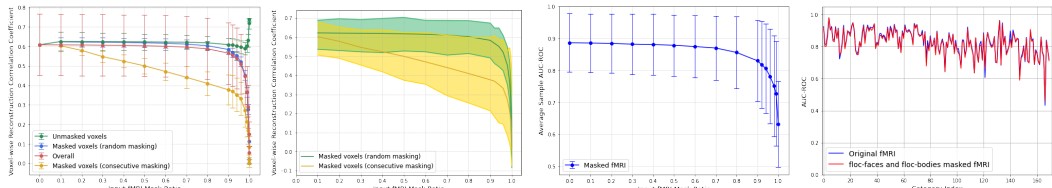

(a) fMRI reconstruction performance.  (b) fMRI multi-label classification performance.

Figure 1: Redundancies observed in the fMRI betas in terms of **(a)** voxel activity reconstruction, and **(b)** stimuli category classification. For reconstruction with an autoencoder, we gradually increase the masking ratio of the inputs. Two masking schemes are tested: randomly choosing masked voxels and consecutively masking voxels. With random masking, the reconstruction performance stays around the same level up to masking 80%-90% of the voxels. The right plot of (a) shows the min, mean, and max reconstruction performance of the two masking schemes. For classification results (b), the left plot shows sample-wise AUC-ROC as we increase the masking ratio: the significant drop also occurs after 80%-90%; the right plot shows the category-wise AUC-ROC with and without masking voxels in the floc-faces and floc-bodies ROIs, which turned out to be extremely close to each other, not affecting *person* category (index 0)'s performance.

Two models are involved in this study: one for voxel activity reconstruction and the other for signal classification. For both models, the inputs are flattened fMRI betas from nsdgeneral voxels. For the reconstruction model, we use an autoencoder (AE) with a three-linear-layer encoder and a three-linear-layer decoder. Nonlinear activations (ReLU) are added between the layers. The hidden dimension of the bottleneck representation is 1024. The reconstruction model is trained with mean squared error loss, and we measure its performance using the voxel-wise reconstruction correlation coefficient. For the multi-label classification task, we train the model to classify if specific object categories exist in the image stimulus that triggers the corresponding fMRI. The category information is obtained from MS-COCO (Lin et al., 2014) as all NSD images are sampled from this image set. In total, there are 171 categories, including 80 bounded "things" categories (e.g., *person, car*) and 91 unbounded "stuff" categories (e.g., *sky, sea*). The classifier consists of 3 linear layers with ReLU activations in between, followed by a Sigmoid activation. The classification model is trained with binary cross-entropy loss, and we measure its performance using AUC-ROC. We perform "masking" on the inputs to remove the information content of voxels. In the context of this paper, masking is done by setting the voxel values to zero while keeping the input dimension unchanged.

## 3 Results

### 3.1 Brain signals contain high-level redundancy.

We studied the performance degradation when increasing the input masking ratio, ranging from zero (unmasked) to one (all-zero inputs). As shown in fig. 1, we observe that both reconstruction and classification models can retain the same level of performance even when masked up to 80%-90% of voxels if the masked positions are selected randomly. The most drastic performance drop happens after masking more than 95% input voxels. Nonetheless, for the reconstruction model, with input activities from only 16 random voxels, the prediction of all the masked voxel activities can still achieve an average of 0.113 voxel-wise correlation coefficient. This indicates that a considerable amount of redundancy exists in the fMRI signals. This redundancy is also more localized: if the masking is performed consecutively, where nearby voxels are masked together, the performance will decrease more consistently. In a consecutively masked window, the two ends also typically observe better performance as they can get information from unmasked neighbors. In addition to randomly masking voxels at different ratios, we also tested masking entire ROIs for the classifier inputs. In particular, we masked all voxels in floc-faces and floc-bodies, and computed the category-wise AUC-ROC. Surprisingly, the performance with original and ROI-masked inputs are almost the same (as in fig. 1b), even for the *person* category. This result suggests that brain signals also carry **hierarchical redundancy** apart from **local dependency**: the model can classify the signal based on activities in the low-level visual cortex without relying on regions with functional specializations. Note that this is different from compression or dimensionality reduction of the signals, since masking directly removes information in the observed signal space instead of in a transformed basis.

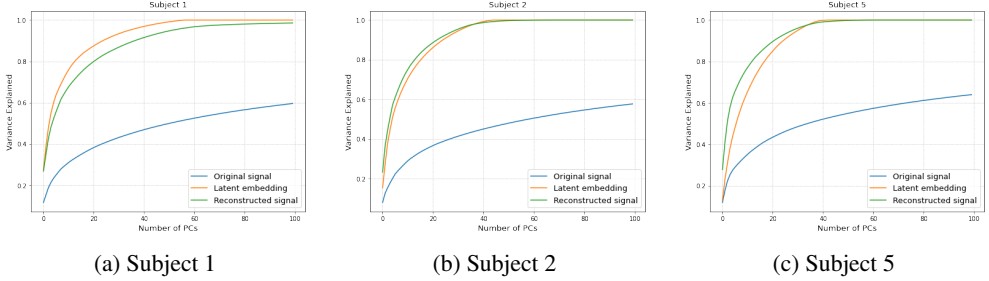

| (a) Subject 1 | (b) Subject 2 | (c) Subject 5 |

Figure 2: Accumulated explained variance v.s. the number of principal components (PCs) used. The trend of the reconstructed signal is close to that of the latent representation.

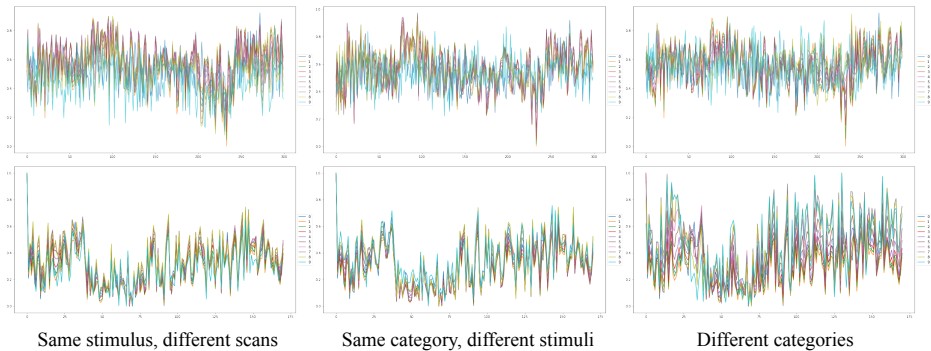

Same stimulus, different scans    Same category, different stimuli    Different categories

Figure 3: Interpolating the latent embedding and generating reconstructions from the interpolations. The top row shows the generated signals for 300 voxels, and the bottom row shows the classification logits for 171 categories when passing the generated signal through the trained multi-label classifier. Values in both plots are normalized to 0-1. Interpolations are performed between three pairs of embeddings: (left) between two fMRIs corresponding to the same image; (middle) between two fMRIs corresponding to different images, but with exactly the same set of object categories; (right) between two fMRIs corresponding to two images having completely two different sets of object categories.

## 3.2  Autoencoders effectively "denoise" brain signals.

Just as Robust Principal Component Analysis (RPCA) can decompose images into low-rank and noise/outlier corruption components (Candès et al., 2011), an AE can "clean up" the high-dimensional signals by reconstructing them into low-dimensional ones. When applying PCA to subject 1's 1000 samples (15724 voxels each) in the original signal space, 100 principal components (PCs) can explain 0.596 variance. Brouwer & Heeger (2009) also mentions that the number of PCs required to explain 0.68 variance is typically two orders of magnitude smaller than the original number of voxels, meaning the number of PCs required for the 15724 voxels is on a hundred-scale. On the other hand, 100 PCs can explain more than 0.99 variance for both latent representations and reconstructed signals (fig. 2). This shows applying an AE can effectively obtain a cleaner version of fMRI signals. In comparison, we found neither RPCA nor independent component analysis (ICA) can achieve the same level of compression. To ensure that our reconstructions retain the primary information, we compared the model's voxel-wise reconstruction correlation with the voxel noise ceilings and found a good alignment between the two (see appendix fig. 5 for details). After adjusting voxel-wise performance with their noise ceilings, there is uneven performance across the brain: we observed better reconstructions for voxels in the higher-level regions.

## 3.3  Brain activity resides in the semantic space—the Hopfieldian view.

We further studied the latent representations obtained from the encoder. Given two fMRI signals, we interpolated their latent embeddings and passed the interpolations through the decoder to generate a set of fake fMRI signals. We then used these generated signals as inputs to the classifier model and obtain predicted logits for each class. As shown in fig. 3, the generated signals in the voxel space differ much more than their predicted logits: this is especially true for interpolations between the same image's different scans or two scans corresponding to images with a similar semantics. This result suggests that brain signals naturally reside in a more semantically rich space. Therefore, our findings support the recent claim that the Hopfieldian view, where representations and transformations

in the neural space are considered more important than individual neuron activities, is needed to explain cognitive processes (Barack & Krakauer, 2021). Appendix fig. 6 discusses the scenario when the interpolation happens in the original signal space instead of the latent embedding space: we found that brain signals can have rich representations with easy transitions. Moreover, we observe that voxel groups affect the latent representation in a consistent manner; for example, masking voxels in floc-bodies always results in representations further away from the original embedding than masking floc-faces voxels, indicating the former ones are more important to the overall latent representation (results are shown in appendix fig. 7).

### 3.4    Masking and attribution reveal voxel and region importance.

By masking part of the inputs, we can investigate which area has more information by comparing the degradation of the reconstruction results. This section aims to understand the differences among different categories, hemispheres, ROIs, and general redundancy patterns through masking, providing constrained inputs, and other input attribution methods.

**Categories.** A natural hypothesis is that the brain encodes various categories with different levels of redundancy: this can be either a "nature" phenomenon shared by the population or a "nurture" one influenced by individual experiences. To test this, we separated fMRI signals based on the categories of their corresponding stimuli. For a stimulus with multiple objects from different categories, we include its fMRI signal in all these categories. For unmasked inputs, the reconstruction performance for a category's fMRI follows its occurrence: a category with more fMRI samples has a higher chance of having its fMRIs better reconstructed. However, with partially masked inputs, the category performance orderings differ from the unmasked ordering and remain mostly consistent across masking ratios. These orders are also more relevant to the category semantics (categories that belong to the same supercategory tend to have a clustered performance) and less relevant to the occurrence frequency (see appendix figs. 11, 9, 10 for visual results). Noticeable groupings of categories can be observed based on their performance curve across masking ratios. There are some consistencies across the three test subjects: sports-image triggered signals typically receive better reconstruction, and animal-image triggered ones have the opposite trend. Other orderings are more subject-specific.

One interesting observation is that the brain can learn representations from other categories under the same supercategory. For example, keeping the number of total training/validation samples the same, when we take out the entire *food* supercategory from the training data, the reconstruction of *sandwich* signals ranks 45th out of the 171 categories. But when the training data has other *food* categories, but no *sandwich*, the reconstruction of *sandwich* signals ranks 20th (if there are *sandwich* signals in the training data, its reconstruction ranks 9th). Taking out an entire supercategory also makes the overall reconstruction worse for all categories. These show that signals of each supercategory have a relatively unique representation. We also tested dichotomous separations, where signals are separated based upon a single category (e.g., *person* and *non-person* triggered fMRIs). Upon testing several partitions based upon *person, tree, sea, cow, building*, we found that only *person/non-person*'s performance has a systematic difference across subjects: person fMRIs are reconstructed better than non-person fMRIs in low-to-mid-level visual regions of the right hemisphere.

**Hemispheres.** There is a known asymmetry between the two hemispheres when it comes to visual processing: from the early study of split-brain patients (Gazzaniga, 1970) to later experiments with hierarchical letters and objects (Sergent, 1982; Christie et al., 2012), results suggest that the right hemisphere (*rh*) is better at identifying a global feature while the left hemisphere (*lh*) is better at processing local ones. But *which hemisphere's information can be better recovered from other regions when damaged?* Masking an ROI's signals can conveniently emulate a lesion in an experimental setting. In our study, we masked 300 voxels in each of V1 to V4 for either *rh* or *lh*, and measured the reconstruction performance for both masked and unmasked regions. Interestingly, we observe no change in the performance of unmasked voxels but statistically significant differences, with t-test $p < 1e^{-8}$, between $lh/rh-$V3/4 for masked ones. Across subjects, $rh-$V3/4 can be better reconstructed from other voxels activities (shown in appendix fig. 8), suggesting they are more robust to region damages than the $lh$ counterparts. Moreover, since the separation of "local" and "global" can be dynamic and depends on the task, as Robertson & Ivry (2000) suggests, we evaluated if the same phenomenon occurs for reconstruction. To this end, we tested fMRI signals corresponding to different categories (the categorical signal separation is the same as in the previous section). However, no noticeable performance difference is observed: we compared the reconstruction performance of

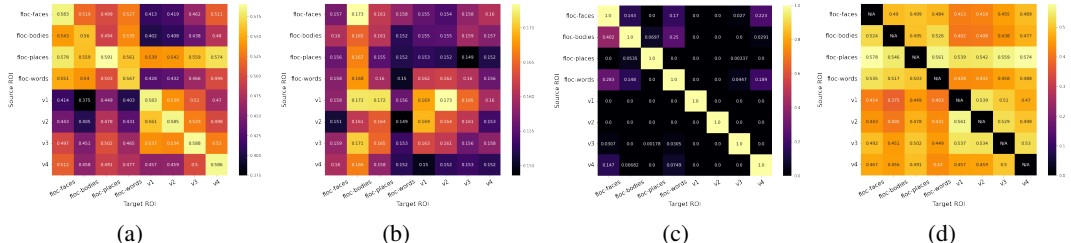

| (a) | (b) | (c) | (d) |

Figure 4: Pairwise reconstruction between ROIs of subject 1. For each matrix, the rows are source ROIs that provide input activities, and the columns are target ROIs whose reconstruction performance is evaluated (in terms of the voxel-wise correlation coefficient). The plots are: **(a)** reconstruction mean, **(b)** reconstruction standard deviation, **(c)** voxel overlap percentage normalized by the total number of target ROI voxels, and **(d)** reconstruction mean when the overlapped voxels' activities are not provided in the inputs.

the following categories, *person, tree, sea, kite, book, cow, giraffe, cat, dog*, and only *cow* differs from the others or the average.

**Pairwise ROIs.** By providing an ROI A's activities as inputs, we can evaluate the reconstruction of another ROI B to understand brain region dependencies. We measured pairwise reconstructions between four ROIs V1-V4 of the visual cortex and four functional localizer (floc) ROIs. The result provides a straightforward dependency matrix as shown in the fig. 4. In general, given voxels from floc ROIs, voxels in higher visual areas are reconstructed better, confirming the high-level nature of floc ROIs. In particular, floc-places can recover visual cortex voxels much better, implying that the high-level place representations cover visual details more than other tasks. On the other hand, given visual cortex voxels, floc-faces and floc-places are reconstructed better, indicating the other two tasks require additional information from other regions apart from the visual cortex. Note that even for self-reconstructions, where the target ROI is the source ROI, the reconstruction performances differ, suggesting that some ROIs are more self-contained than others and their dependencies are more local. We also conducted studies between different visual cortices; see appendix fig. 12 for more details.

**Optimal voxel measurement for reconstruction and classification.** Considering that low-dimensional embeddings can effectively represent neural signals, a reasonable question is whether one can utilize techniques similar to compressed sensing and use fewer measurements to obtain high-resolution samples. Here we explore the dependencies between voxels to find voxels that (1) contribute more to other voxels' reconstruction and (2) contribute more to the semantic categorization of the signals. For (1), we chose a target voxel to measure the reconstruction performance while providing voxels from V1-V4 plus one additional voxel. Then, we brute-forced this additional voxel and plotted a dependency heatmap. We observed strong local dependencies and dependencies in the symmetric positions of the other hemisphere. When the "contribution" is aggregated for all target voxels, we can get a blueprint regarding which voxel is most important. However, this result is subject-specific, and more explorations are needed to extend the results to different individuals and achieve a better upsampling of the signals. For (2), we utilized SHapley Additive exPlanations (SHAP) (Lundberg & Lee, 2017) to perform input attributions, identifying voxels' contribution to classifying each category. Aggregating the attributions for all the categories results in one over-all contribution value for each voxel, thus providing a general voxel importance map for signal categorization. (appendix fig. 13 has more details for both cases).

## 4 Conclusion

In this work, we systematically studied the redundancy and dependency of fMRI signals with an AE and a multi-label classifier. We found AE can reconstruct signals in a much lower-dimensional space while having a good reconstruction correlation: this suggests new ways for signal decomposition and denoising. We also found signals' latent space is more aligned with the semantic space, supporting a Hopfieldian view of the brain. This low-dimensional representation, or semantic information of stimuli, can be further used to guide signal compression, upsampling, and reconstruction. In addition, our results suggest that the brain encodes different scene semantics with varying levels of redundancy, resulting from a combination of nature and nurture. Discrepancies between hemispheres, ROI dependencies, and voxels dependencies are also explored, each providing additional insights regarding brain visual encodings.

## Acknowledgments and Disclosure of Funding

This project was partially supported by National Science Foundation under IIS-1817046 and HDR DSC 1924205.

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

# A Appendix

## A.1 Experiment details

Since each image stimulus in NSD corresponds up to three scans, we split our training-validation data image-wise: for example, for subject 1, 23715 samples corresponding to 8364 images are used as the train set, and 4035 samples corresponding to the remaining 1477 images are used as the validation set. Therefore, our pipeline never sees the image it is tested on during the training. We use $Adam$ as the optimizer, with a 2e-4 learning rate, 2e-4 weight decay for the reconstruction model, and 8e-6 weight decay for the classification model. The values are manually tweaked with grid search. Since input lengths vary across subjects ($N = 15724$ for subject 1, $N = 14278$ for subject 2, and $N = 13039$ for subject 5), models are trained in a per-subject manner. Our experiments are conducted on a Tesla V100 GPU with 32 GB memory (however, only around 4K MB memory is needed for a batch size of 64). The code provided in the supplementary material contains the exact model parameter settings (hidden dimensions) we used.

## A.2 Limitations and future work

Although we studied many aspects regarding brain signal redundancy and dependency, the list is by no means exhaustive. For example, one can examine the impact of the dorsal/ventral stream on fMRI signals of different categories. Other questions can also be explored: the brain can encode both low-level and semantic information upon seeing images, so *how much of each (low-level/semantic) is retained in the latent embeddings when the fMRI signals are mapped from the encoder?* To answer this question, we will need to train a model trained from latent embedding to perform multi-label classification, and compare the performance with the model trained with fMRI inputs. Similarly, we need to train two other models for low-level details: one takes fMRI signals as inputs and another from the latent embeddings. Then we can compare the retainment percentage of low-level image information. These latter two models require labels of the stimuli regarding image details, such as color, shape, orientation, etc., which are missing in the current MS COCO dataset. In future work, one could potentially use off-the-shelf detectors to generate pseudo labels as a way to answer the above question.

In addition, since linear layers have less inductive bias than convolutional layers, the models we used are composed of linear layers. In fact, we also tested a convolution-based autoencoder VQ-VAE (Van Den Oord et al., 2017), as discussed in fig. 13. With strong local-grouping assumptions of the convolution operation, VQ-VAE fails to capture global dependencies. Nonetheless, we still observed similar curves for masked voxel reconstruction: the noticeable performance decrease starts after the 80% masking ratio. Its results on category reconstruction orders, latent embeddings' semantic alignment, orders of pairwise ROI dependencies, etc., are also consistent with the linear model results we reported. However, it will still be beneficial to test models with more structural types in future work to examine if the results hold for all frameworks.

## A.3 Figures

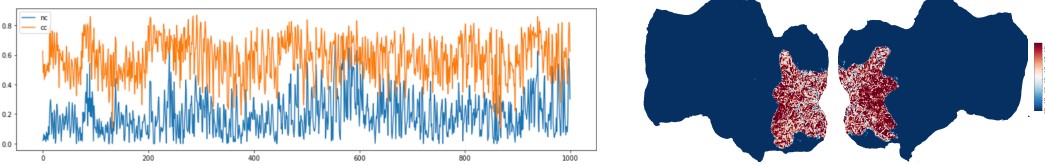

Figure 5: The left plot shows the reconstruction correlation coefficient ($cc$) together with the voxel-wise noise ceiling ($nc$) for 1000 voxels. $cc$ is calculated over the validation set of 4035 samples and aligns well with $nc$ ($cc$ and $nc$ have a 0.67 correlation with p-value 0). The right plot shows $cc - nc$ values on a flatmap. Higher-order regions typically have larger values (redder), meaning the reconstruction is better for those regions.

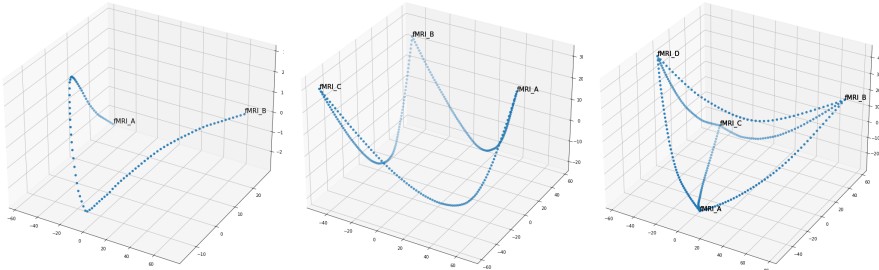

Figure 6: Latent representations of interpolated fMRI signals between two fMRI samples, pairs in three samples, and pairs in four samples. The interpolations occupy a much lower dimensional space: for 1000 interpolated latent representations between two fMRI signals, only 2 PCs are needed to explain 0.998 variance, whereas 56 PCs are required to explain the same variance for 1000 unrelated fMRI signal embeddings. This indicates transitions between different fMRIs are cheap inside this learned latent space. For visualization, the first three PCs of the latent representations (>0.999 variance explained for each interpolation pair with step number = 1000) are used as the coordinates.

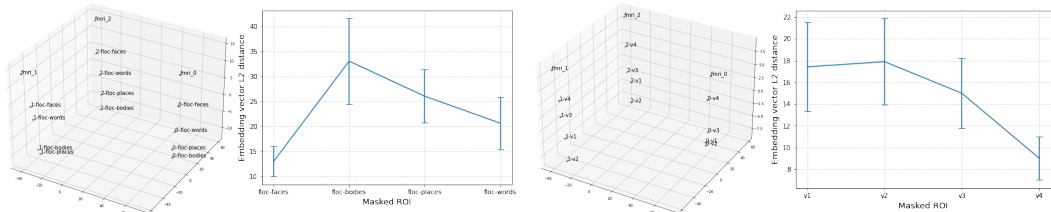

Figure 7: Masked fMRI embeddings. Masking different ROIs results in different distances between the masked and original signal embeddings. The relationship between these distances is consistent across samples (all pairs have a t-test p-value < 1e-50 except for the V2>V1 pair, which has a p-value of 0.008): for example, masking floc-faces voxels always results in a closer embedding to the original signal to masking floc-bodies voxels. These distance relationships hold across subjects.

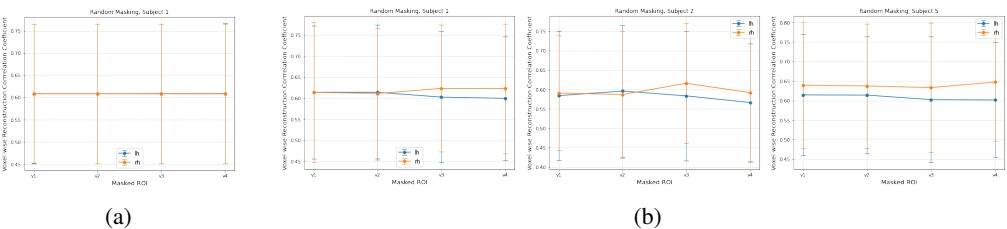

Figure 8: (a) Unmasked voxel and (b) Masked voxel reconstruction performance when masking V1-V4 on either the left hemisphere (*lh*) or the right hemisphere (*rh*). Subjects 1, 2, and 5 are used for the task. Masking the visual cortex at different levels on either hemisphere does not affect unmasked voxels differently (true for all subjects). But the reconstructions of masked V3/V4 voxels on *lh* consistently have a worse reconstruction than those on *rh* (t-test p-value < 1e-8).

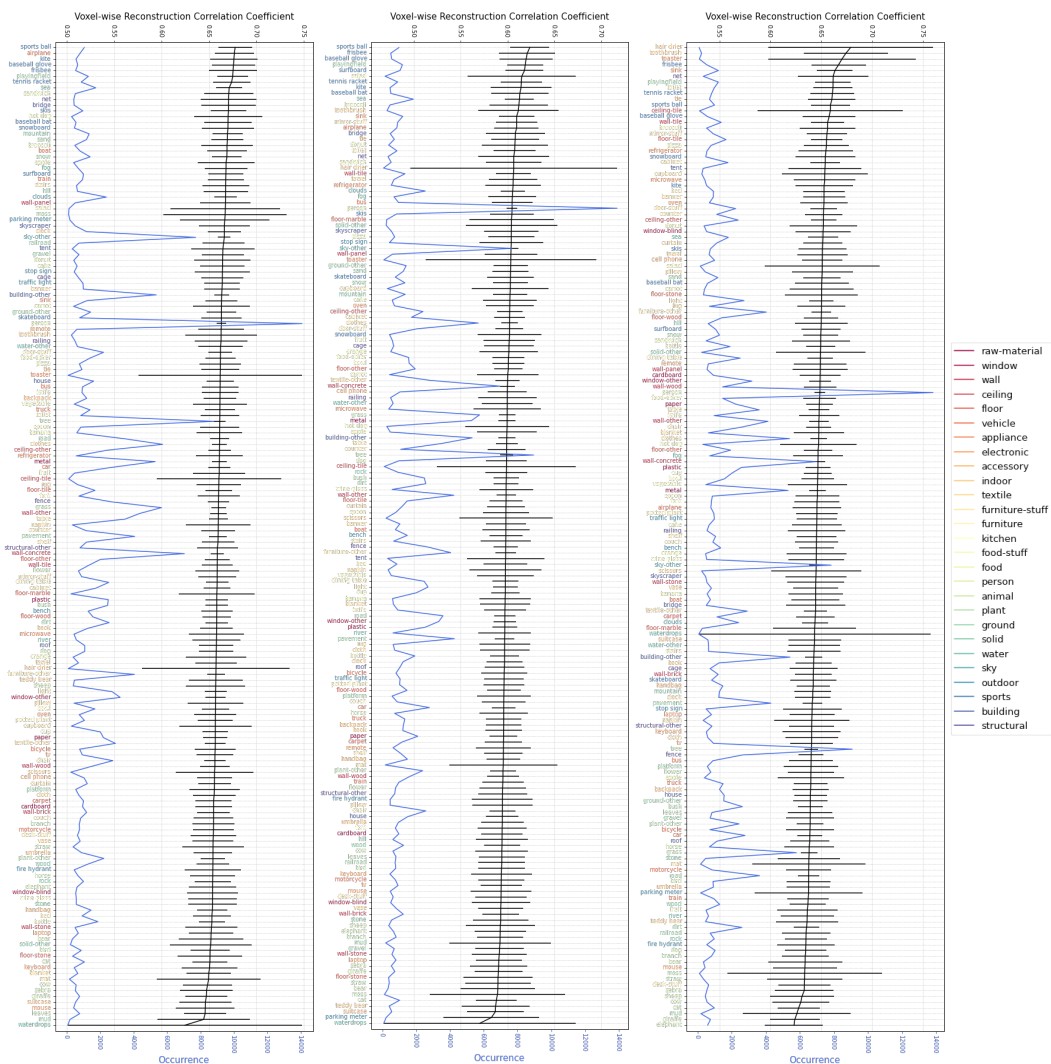

Figure 9: Average voxel-wise reconstruction correlation coefficient (*cc*) for fMRI samples corresponding to different categories. The performance is measured under a 50%-masking ratio. Plots are for subjects 1, 2, and 5 from left to right. Error bars stand for the standard deviations of the average *cc* across all samples of that category. We also plot the sample occurrence (in blue lines) for individual categories as some categories, like *person*, have significantly more samples than other categories. Colors are based on super-categories, as indicated in the legend.

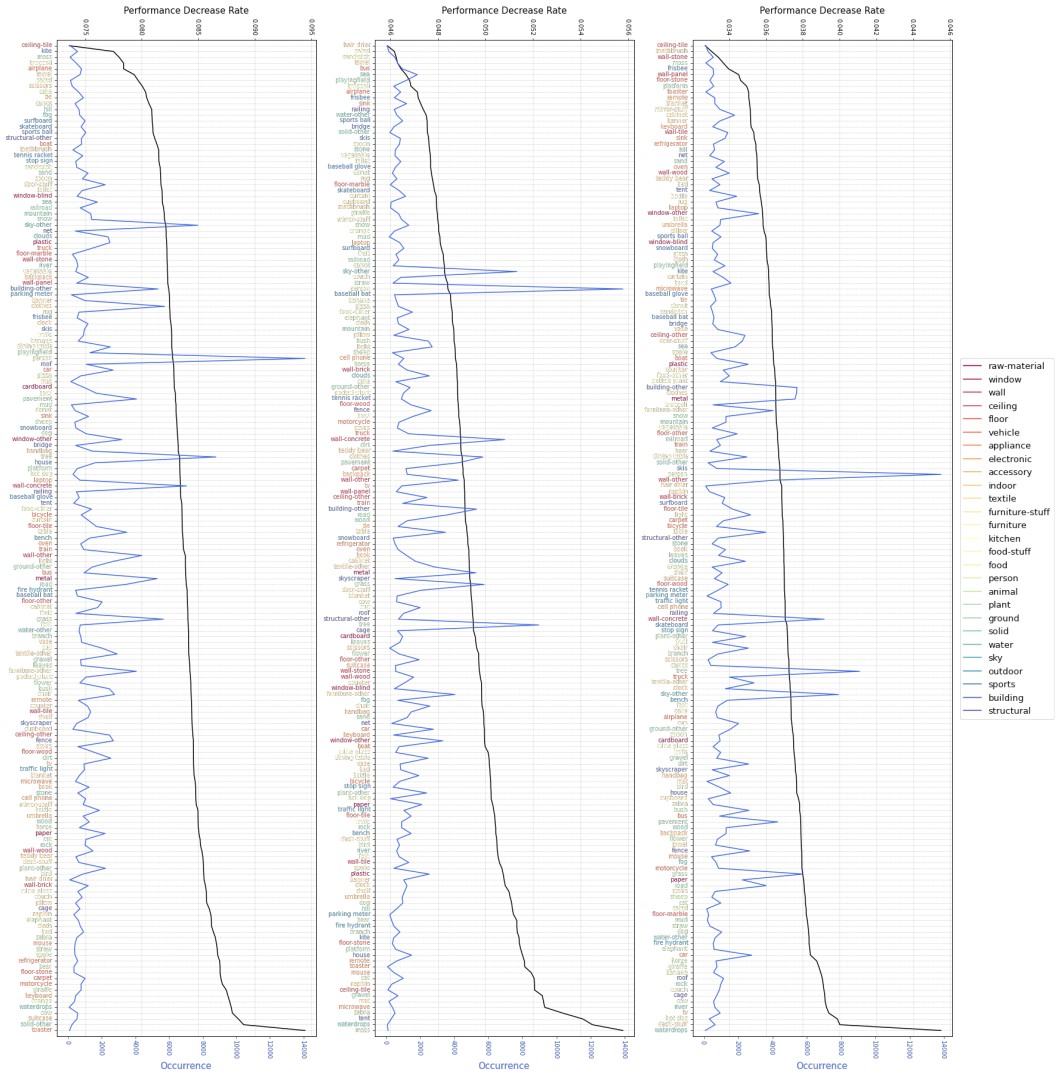

Figure 10: Performance decrease rate for different categories, together with the category sample occurrence. The rate is calculated as (performance with 10%-masked inputs - performance with 90%-masked inputs) / performance with 10%-masked inputs.

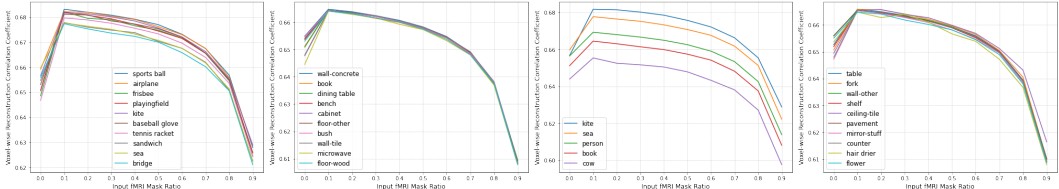

(a) Reconstruction performance of different categories under different masking ratios.

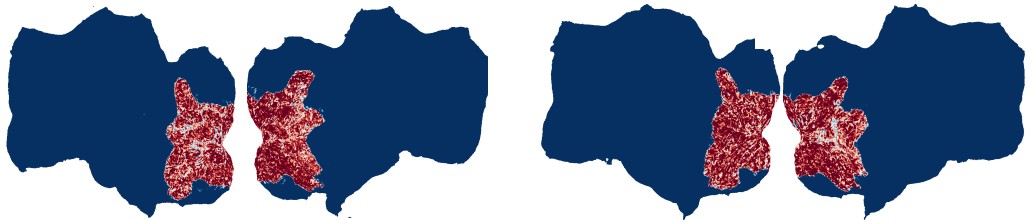

(b) Reconstruction performance of *person* fMRI minus that of *non-perso*n fMRI (L: subject 1; R: subject 5).

Figure 11: Reconstruction for different categories **(a)** From left to right: (1) categories exhibit groupings, (2) unmasked order is very different compared with masked ones, (3) the orders are consistent for categories in different groups for masked inputs, and (4) the orders can be inconsistent for categories within the same group (as in having a close performance under the 10%-masking ratio). **(b)** fMRI signals triggered by images containing *person* perform better (redder) at the right hemisphere's low to mid-level visual regions than those that do not contain *person*. In addition, when calculating the correlation between *cc* and *nc* (refer to fig. 5), person ones are larger than non-person ones (with an average of 0.673 v.s. 0.638 for subject 1).

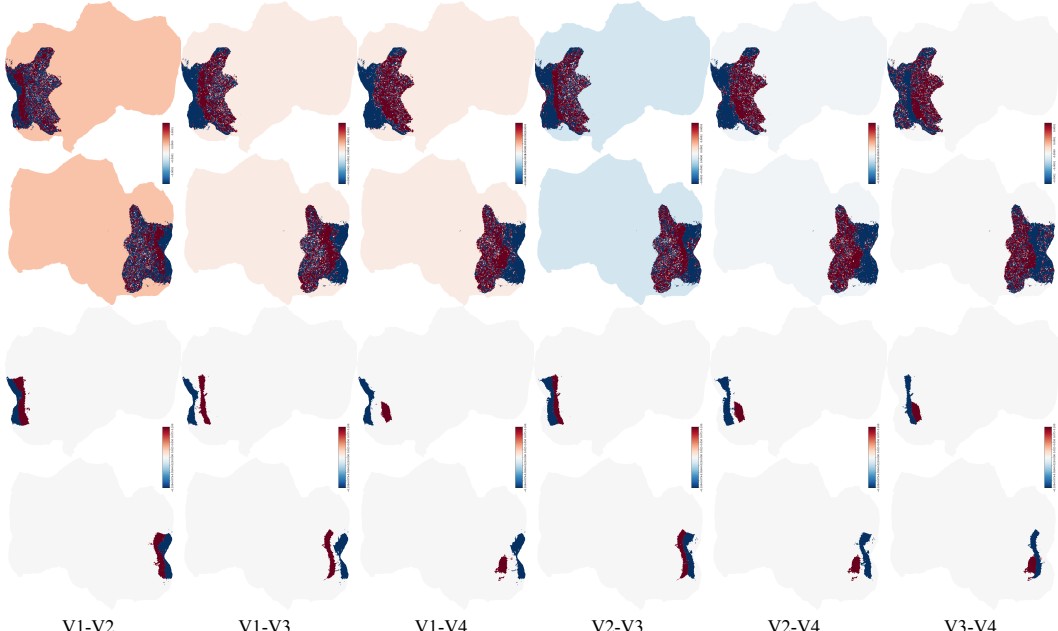

Figure 12: The top two rows of the flatmaps are reconstruction performance differences when masking two different visual cortex ROIs. The bottom two rows are visualized localizers of corresponding ROIs. Both sets have *rh* on top of *lh*. The region name stands for the masked region: for example, the first column "V1-V2" means "subtracting the voxel-wise reconstruction with V2-masked inputs from the voxel-wise reconstruction with V1-masked inputs". Discrepancies are observed between the performance difference (top set) and corresponding localizers (bottom set), from which we can identify region dependencies. For example, V2 depends more on V1 than V3, and it also shows additional dependencies with the posterior intraparietal sulcus (IPS) area on the right hemisphere (top tip of the rh flatmap). Together with the pairwise ROI analysis under section 3.4, these experiments that aim to find dependencies between brain regions can also link to works in connective field modeling (Haak et al., 2013; Knapen, 2021).

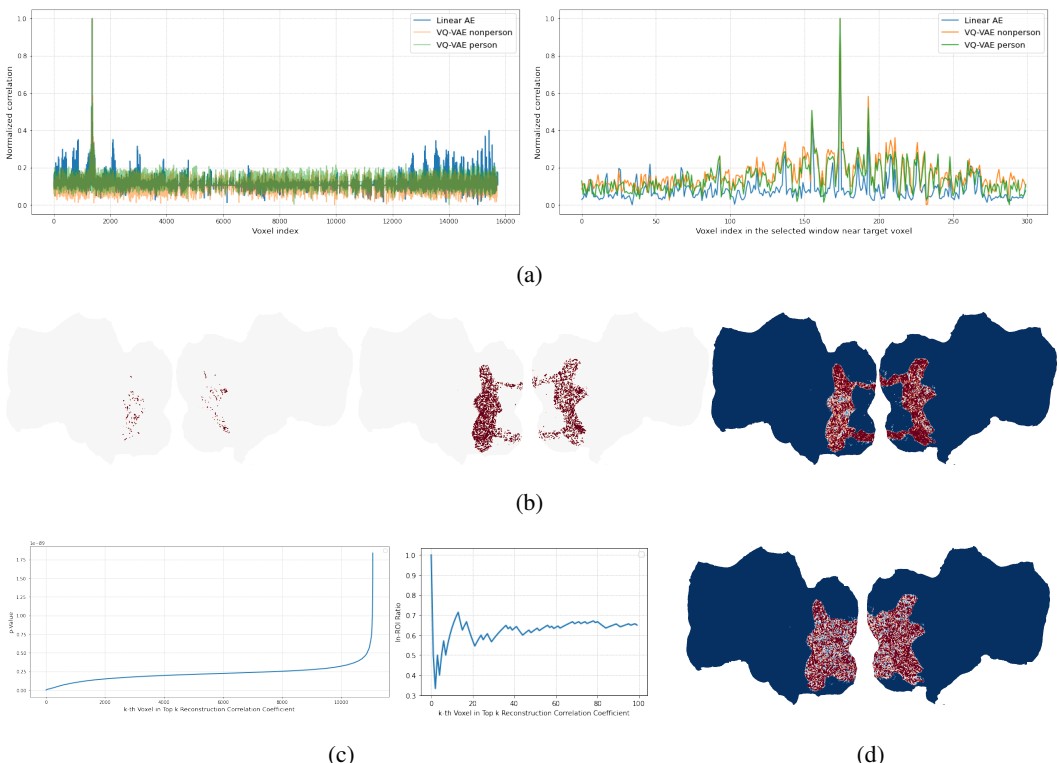

(a)

(b)

(c)                                        (d)

Figure 13: Examining the voxel importance of subject 1 with (a-c) reconstruction models and (d) the classification model. For the reconstruction, we tested the AE detailed in the main text, as well as an additional VQ-VAE with a convolutional encoder and decoder and a hidden dimension of 984. Voxel with index 1364 is selected as the target voxel at which we measure the recovery performance; it locates on $lh$ and belongs to both floc-faces and floc-bodies. Voxel activities in V1-V4 are always provided in the inputs, along with an additional voxel activity. **(a)** The recovery performance at each voxel if they are served as this "additional" input. The right plot is the zoom-in view of the left plot, showing the strong local dependencies, which are consistent across models and signal types. Compared to linear AE, convolution-based VQ-VAE can only capture local dependencies well but lose the global view. Nonetheless, it can reconstruct unmasked signals much better, with $cc$ having a mean of 0.968 and 0.003 std (the values in (a) are normalized to 0-1 since AE and VQ-VAE $cc$ are at different scales). **(b)** Given AE's result, we plot (left) the top 200 contributing voxels, (middle) voxels that lead to a reconstruction performance larger than the mean value, and (right) the overall p-values of the reconstruction $cc$. We can observe that voxels on mirrored positions of $rh$ are also contributing to the target voxel's reconstruction, but overall $lh$ voxels are more important for this target voxel on $lh$. p-Values are also aligned well: positions with better performance (larger $cc$) also have smaller p-values. **(c)** (left) The p-value changes with decreasing recovery $cc$. There is a sharp increase near the end, indicating those voxels are more irrelevant; (right) The in-ROI ratio for the top-100 contributing voxels: not all are from the ROIs that the target voxel belongs to (floc-faces/bodies in this case). **(d)** SHAP input attribution (absolute values) aggregated across categories. Redder (higher attribution) means the voxel is more crucial in determining if a specific category exists in the stimulus. There is a disparity between the two hemispheres, with $rh$ having more critical voxels in higher-level regions. However, the attribution of $lh$ is generally higher than $rh$ (t-test p-value smaller than 0.005). When looking at (unaggregated) category-wise SHAP attributions, the results also support distributed coding scheme (Thorpe, 1989): each category's representation is coded by a subpopulation of voxels, and each voxel contributes to multiple categories' representations.