# OpenReview forum: "Redundancy and dependency in brain activities"
_NeurIPS.cc/2022/Workshop/SVRHM — SVRHM Poster_

### Official Review · Reviewer_2B69 · 2022-10-04
**Interesting results - suitable for this workshop in terms of exploring human representations of visual information**

**Rating:** 6
**Confidence:** 4

**Review:**

Interesting direction, this paper falls under the workshop’s aim to identify relevant representations of human visual perception in terms of quantifying redundancy in fMRI signals .

Summary

This paper probed redundancy in fMRI signals by testing the ability of signal reconstruction and classification of voxel activities. They used a VAE model for signal reconstruction and a classification model comprised of linear layers to classify visual stimuli, both using flattened fMRI signals as the input. They masked input voxels both randomly and consecutively. They found that signal reconstruction and classification is still feasible with 80-90% of voxels randomly masked.

Pros

Very cool research direction that does seem to directly support Hopfieldian views of cognition and may give insight into how current computer vision models, which have been found to have high levels of redundancy, may still be biologically similar to the ventral stream in this way.

Thorough experimentation was performed in terms of investigating different brain regions and stimuli, and brain regions and voxels.

Cons

Why not also provide either class accuracy or mean-class accuracy in addition to AUC. It is hard to interpret how well the classifier model is performing especially since this is an 80 class problem. It seems surprising that all 80 classes are still readily classified after 80% of fMRI signal is masked.

Need to introduce terms such as fMRI-betas, floc-faces and floc-bodies in a more detailed context.

All figures are too much small to read the text and labels

Minor

Why not use all 8 subjects available in NSD?

Limitations should be in main text and not as an appendix note

---

### Official Review · Reviewer_nnk5 · 2022-10-14
**Interesting methodology but analyses can be improved**

**Rating:** 6
**Confidence:** 3

**Review:**

This work studies redundancy and dependency in fMRI measurements of the brain activities. The main methodology involves using autoencoders (AE) for reconstruction and multi-label classification models applied to fMRI signals. Although the methodology is interesting and the study shows how it can be used for various aspects, there are some questions as well.

- On hierarchical redundancy (line 68): If you just include voxels in floc-faces and floc-bodies what would be the performance? This would give a better idea how strong the signal relevant to faces or person is in the corresponding area, and thus whether a significant drop is expected when voxels in those areas are masked.

- On denoising effect of AE (section 3.2): The observation that AE can reconstruct signals in a low-dimensional space is interesting, effectively leading to denoising effect. How is the effect affected by the choice of AE model architecture? In particular, if we use a higher latent dimension, do you expect to see less denoising effect? Some discussion on whether you would almost always see denoising effect or need to choose the right architecture would be interesting.

- On Hopfieldian view (section 3.3 and Figure 3): The comparison between the predicted logits and fMRI signals is qualitative. Can you add quantitative measures for comparison? Also, what is the justification for comparing the logit outputs from the network and the brain activation? It is probably not surprising that the brain representation is far from some probability distribution across different categories as the brain is not optimized just for visual classification. Another question is whether the logit activations are better aligned with the brain activation after a linear transformation.

- Line 125: The description “the brain can learn” is misleading— it is an artificial network that is being trained not the biological brain.

- Category-depending masking (section 3.4): The analysis is rather anecdotal. More systematic and quantitive analyses would be helpful.
Pairwise ROI reconstruction: I find this experiment interesting, but Figure 12 is hard to interpret.

- Overall, I feel like writing and presentation can be improved. While various analyses have been provided, I think the motivation behind them and their significance can be emphasized more. Also, more quantitive analyses in addition to qualitative ones would be desirable.

---

### Official Review · Reviewer_7PSe · 2022-10-14
**The overall aim of the paper comes across in a clear way. The presentation of the findings however, can be improved in terms of coherence. and clarity.**

**Rating:** 7
**Confidence:** 3

**Review:**

Quality
The quality of this paper is good in my opinion. The way however the results are presented can be improved by compressing the figures and supplementary information (see detailed comments below) such that the story becomes more coherent and convincing.

Clarity
The introduction is written in a clear way such that it is apparent for the reader what the goal of the project is as well as previous work on which it is based.

The paper presents a lot of results, both in the main text as in the supplementary material. Due to the fact that in the text there are often results discussed by referencing figures in the appendix, the overall structure sometimes feels a bit incoherent. It’s also unclear to me why some figures are placed in the main and others in the supplementary text. For instance, Section 3.4 - Categories, discusses results by referencing 3 figures in the appendix. It takes quite some effort to understand how these figures show the discussed results, making the main point come across less well. Another example is Section 3.4 - Pairwise ROIs, where the claims in the text are illustrated with Figure 4. There is however, no explanation what the four different matrices represent in the main text, and how these numbers lead to the proposed conclusions.

In the figures the axes are often really small up to the point that they become unreadable (e.g. Figure 3 - legend and ticks - and Figure 4 - correlation values). Also, the relevance of some of the figures was unclear to me, e.g. Fig. 4: several of the matrices show information which is not discussed in the text. Also the right plot in Figure 1a is also not discussed and it was also unclear to me what new information it shows in addition to the left figure.

Originality & Significance of this work
The approach where coefficients of a GLM model are used in combination with an AE seems like an interesting way to test the redundancy and dependency of brain activities. The combination of activity prediction as well as object classification allows for an interesting analysis of both the model and the information contained in the voxel activities.

Pros
The variety of analysis performed gives a lot of information about the redundancy and dependency of brain activities.

Cons
The figures are often difficult to read because the font sizes are quite small. The quality of the paper could be improved by an overall larger font size, especially for  Figure 3 where the ticks and legends are unreadable and Figure 4 where the correlations in the matrix are also almost unreadable.

Other comments
Motivation for only using the data from subject 1, 2 and 5 was unclear to me.

(Pg. 3) - Section 3.2: it was unclear to me what exactly was meant with a sample (also after reading the information given in the appendix).

---

### Official Review · Reviewer_8cCt · 2022-10-17
**Redundancy and dependency in brain activities**

**Rating:** 6
**Confidence:** 4

**Review:**

The manuscript provides evidence fMRI-based evidence for an account of redundancy and dependency of information signals in the brain. I believe the decoding/classification methods and results presented here to be well explained. These results will also be of great interest for future investigations on neural redundancy especially in the space of neuroimaging/human neuroscience as they provide a great foundation and framework on this topic. With that said, I have a few comments that could be helpful for the authors in further improving the manuscript.
1.	Most, if not all, figures are very challenging to see. In most cases, it is impossible to tell what different components of each subplot represents and the figure captions are not always sufficiently descriptive.
2.	It would be helpful if the authors could include more information on the exact experimental setup for the fMRI data collection and preprocessing pipeline. While this seems to be of little relevance to the main analyses of the manuscript, different choices of preprocessing and parameter optimization prior to classification/decoding could drastically change the observed results. This information would also be helpful for other fMRI researchers who might be interested in employing similar methods to what proposed in the presented manuscript and as it stands, there is not enough information to allow this. Relatedly, given the poor temporal resolution of fMRI signals, it would be helpful if the authors could explicitly discuss limitations to be considered when interpreting the observed results.
3.	It is unclear why specific reconstruction methods were chosen as the main analyses. It would be helpful if the author include more information on the background literature of the current state of fMRI decoding/reconstruction analyses and how/why the proposed method is a better (or comparable) approach. It would be helpful to understand what aspects of the fMRI signals are in fact redundant and what (if any) are unique across voxels and how these are differently captured in your reconstruction method (as compared to others e.g., inverted encoding models).